# Dietary Black Soldier Fly (*Hermetia illucens*)—Dipterose-BSF—Enhanced Zebrafish Innate Immunity Gene Expression and Resistance to *Edwardsiella tarda* Infection

**DOI:** 10.3390/insects15050326

**Published:** 2024-05-01

**Authors:** Haruki Nishiguchi, Ibnu Bangkit Bioshina Suryadi, Muhammad Fariz Zahir Ali, Chiemi Miura, Takeshi Miura

**Affiliations:** 1Graduate School of Agriculture, Ehime University, 3-5-7 Tarumi, Matsuyama 790-8566, Japan; haruwest1341@gmail.com (H.N.); muhammad.fariz.zahir.ali@brin.go.id (M.F.Z.A.); chiemi8@agr.ehime-u.ac.jp (C.M.); 2The United Graduate School of Agricultural Sciences, Ehime University, 3-5-7 Tarumi, Matsuyama 790-8566, Japan; ibnu.bangkit@unpad.ac.id; 3Department of Fisheries, Faculty of Fisheries and Marine Sciences, Universitas Padjadjaran, Jalan Ir. Soekarno KM 21, Sumedang 45363, Indonesia; 4Research Center for Marine and Land Bio Industry, National Research and Innovation Agency, Jl. Raya Senggigi, Kodek Bay, Pemenang 83352, Indonesia

**Keywords:** polysaccharide, dipterose, black soldier fly, immunostimulant, zebrafish, immune response

## Abstract

**Simple Summary:**

Black soldier fly (BSF) has considerable potential in the aquaculture industry, not only as a protein source but also as an immunomodulator (a substance that can stimulate or suppress the immune system of an organism). This research is the first to investigate a dietary polysaccharide derived from BSF, called dipterose-BSF, and its potential effect on the immune system of zebrafish. Groups of zebrafish fed on a dipterose-BSF diet were compared with a control group fed on a diet with no dipterose-BSF. Fish in the dipterose-BSF diet group showed increased expression of immune-related genes and decreased expression of stress-related genes compared with the fish in the control group. The optimal concentration of dietary dipterose-BSF needed to enhance the immune status of zebrafish was 0.1 µg/g. A higher concentration (1 µg/g) had no significant effect on the immune system of fish in the dipterose-BSF diet group compared with fish in the control group. Hence, the inclusion of dipterose-BSF in the manufacture of aquafeed should be considered.

**Abstract:**

Dietary management using immunostimulants to protect fish health and prevent bacterial infection is widely practiced. Many insect species possess various bioactive substances that can improve animal health. We previously identified several bioactive polysaccharides derived from insects, including dipterose-BSF from black soldier fly (*Hermetia illucens*) larvae; this can stimulate innate immunity in mammalian macrophage RAW264.7 cells. However, the effect of dietary dipterose-BSF on the immune system of teleosts remains unclear. Here, we analyzed the immune status of zebrafish (*Danio rerio*) after 14 days of dietary inclusion of dipterose-BSF (0.01, 0.1, and 1 µg/g), followed by an immersion challenge using *Edwardsiella tarda*. To identify changes in the transcriptional profile induced by dipterose-BSF, we performed RNA-sequencing analyses of the liver and intestine. Differentially expressed genes were investigated, with both organs showing several upregulated genes, dominated by nuclear factor and tumor necrosis factor family genes. Gene Ontology analysis revealed several terms were significantly higher in the experimental group compared with the control group. Challenge tests suggested that dietary dipterose-BSF had some positive effects on disease resistance in fish, but these effects were not pronounced.

## 1. Introduction

The most notable and widely farmed insect for animal feed production is the black soldier fly (BSF; *Hermetia illucens*); it is a non-pest insect [1] and has been widely utilized as a bio-convertor for organic wastes [2,3,4]. BSF larvae contain bioactive substances that can improve animal health; these include antimicrobial peptides (AMPs), which are well-known to be efficient inhibitory compounds against a broad range of pathogens and can be utilized for various purposes [5,6]. Insects can be used in animal feed not only as protein alternatives but also as immunomodulators capable of increasing resistance to diseases [7]. Insects are attracting considerable attention as an animal feed because they are easy to culture, grow rapidly, and they have a short reproductive cycle and a low feed-conversion ratio [8]. Insect meal and its derivatives, such as from silkworm (*Bombyx mori*) [9,10,11], housefly (*Musca domestica*) [12,13,14], cricket (*Gryllus bimaculatus*) [15,16], yellow mealworm (*Tenebrio molitor*) [17,18,19,20], and black soldier fly (*Hermetia illucens*) [21,22,23], have been used in various aquaculture commodities to stimulate an immune response, thus increasing stress and disease resistance.

An interesting finding about insects, especially BSF larvae, is that different extraction methods will yield different substances. Diener et al. (2011) [1] and Harlystiarini et al. (2019) [24] found that BSF larvae extracted using methanol had antibacterial activity against Gram-negative bacteria only, with no activity against Gram-positive bacteria. Diener et al. [1] suggested that these differences in susceptibilities between Gram-negative and -positive bacteria may be due to differences in the interactions between the bioactive substance and the bacterial ribosome or the bacterial cell wall. Ali et al. (2019) [25] found that a polysaccharide obtained from BSF larvae using an ethanol extraction method, which they referred to as “dipterose-BSF”, could induce nitric oxide (NO) activity and various proinflammatory cytokines. This indicated that polysaccharides derived from BSF could also have an immunomodulatory effect.

Polysaccharides are reasonably safe, non-toxic, and biodegradable [26], making them excellent for use in livestock and aquaculture. Thus, the use of bioactive polysaccharides could be an appealing alternative to the use of antibiotics and vaccines in aquaculture [27].

The detection of pathogen-associated molecular patterns (PAMPs) by macrophages in lipids, proteins, peptides, nucleic acids, and carbohydrates can activate innate immunity [28,29]. Furthermore, chitin and chitin derivatives, which are composed of N-acetyl-ß-d-glucosamine and are the main component of the exoskeleton of insects, can promote innate immunity to many pathogens as they are also present in pathogenic fungal cell walls [30] and Gram-positive bacteria [31]. The immunomodulatory actions of polysaccharides are generally highly dependent on structural factors, such as their molecular weight, monosaccharide content, and glycosidic bonds [32,33]. Prior research indicates that polysaccharides with a high molecular mass typically exhibit considerably more biological activities compared with the biological activities of polysaccharides with a low molecular mass [34]. This may be due to the greater number of repetitive structures in polysaccharides of high molecular mass, which could interact with receptors or other membrane targets [35].

Here, we extracted and purified dipterose-BSF, we then investigated its bioactive polysaccharide properties by analyzing the NO level induced by mammalian macrophage RAW264.7 cells as a marker for immunomodulatory activity. We conducted an experiment to investigate the immune responses of zebrafish after they received dipterose-BSF as a dietary supplement for 14 days. This was followed by an *Edwardsiella tarda* challenge test. Furthermore, comprehensive gene expression analysis of zebrafish livers and intestines was conducted using RNA-sequencing (RNA-seq) technology and quantitative real-time polymerase chain reaction (qRT-PCR) analysis. The primary objective of our research was to study the immunomodulatory effects of dipterose-BSF in teleost fish.

## 2. Materials and Methods

### 2.1. BSF Larvae Extraction

BSF larvae were obtained from the Research Institute of Environment, Agriculture and Fisheries, Osaka Prefecture, Japan. We used a previously described extraction method [25]. The BSF larvae were autoclaved at 121 °C for 20 min to constrain endogenous enzyme activity; the larvae were then ground into a homogeneous powder. The resulting BSF larvae meal was diluted with ten volumes of ultrapure water and gently shaken at 4 °C for 24 h. The BSF extract was obtained by centrifuging at 10,000× *g* for 1 h, followed by evaporation in a water bath at 50 °C until one-tenth of its original volume remained, to obtain the concentrated extract.

### 2.2. Dipterose-BSF Isolation and Purification

Dipterose-BSF was purified according to previously reported methods [25,36,37,38]. To precipitate the dipterose-BSF, 99.5% (*v*/*v*) ethanol was added to the BSF extract and gently shaken overnight at 4 °C. The dipterose-BSF was precipitated by centrifugation, washed three times with 80% ethanol, and air-dried inside a draft chamber. The precipitate was diluted in 20 mM Tris-HCl (pH 8.0), shaken overnight at 4 °C, and centrifuged at 10,000× *g* for 15 min to separate the residual low-molecular-weight substances. Crude dipterose-BSF was obtained once the precipitate had been removed.

Gel-filtration chromatography on a HiPrep 26/10 S-500HR column (GE Healthcare, Chicago, IL, USA) was used for the first stage of the dipterose-BSF purification. The crude dipterose-BSF was loaded into the column, pre-equilibrated with 20 mM Tris-HCl (pH 8.0), and then eluted using the same solution with a flow rate of 1.3 mL/min. The eluted fractions were collected automatically. The sugar content was then evaluated using the phenol–sulfuric acid (PSA) method with glucose as a standard [39] and an evaluation of NO (nitric oxide) activity in macrophage RAW264.7 cells was carried out simultaneously. The PSA- and NO-positive fractions were collected and diluted with the same solution used in the subsequent step.

The positive fractions were then injected into a HiPrep DEAE FF 16/10 column (GE Healthcare, Chicago, IL, USA), pre-equilibrated with 20 mM Tris-HCl (pH 8.0), and eluted at a 2 mL/min flow rate with the same solution. The fractions were collected using a linear gradient of 1 M NaCl of 20%, 40%, and 100% at a 2.0 mL/min flow rate. The eluates were collected automatically, and total sugar and NO assays were performed in the same way as stated earlier. Fractions that exhibited NO activity were pooled and dialyzed using PURELAB ultra-pure water (Elga Veolia, UK) by ten volumes, which was repeated five times. The dialyzed fractions was lyophilized to obtain the dipterose-BSF.

### 2.3. RAW264.7 Cell Culture

The RAW264.7 cells were provided by the RIKEN Cell Bank (RIKEN BioResource Center, Tsukuba, Japan). Minimal essential medium (MEM) (Life Technologies, Grand Island, NE, USA), supplemented with 10% fetal bovine serum, 0.1 mM non-essential amino acids, 100 U/mL penicillin, and 100 g/mL streptomycin, was used to sustain the cells. The cells were maintained at 37 °C in humidified air containing 5% CO_2_.

### 2.4. NO and Sugar Evaluation

The NO production in RAW264.7 cells was measured using a Griess Reagent system kit (Promega, Madison, WI, USA), according to the manufacturer’s instructions. The cells were plated at 1 × 10^6^ cells/well in a 96-well plate, pre-incubated for 90 min, and stimulated for 24 h at 37 °C with various concentrations of dipterose-BSF; 100 ng/mL lipopolysaccharide (LPS) was used as a positive control. The absorbance of the culture medium supernatant was measured at an optical density (OD) of 540 nm using a microplate reader. The quantity of nitrite in the culture media was then determined using an NaNO_2_ standard curve.

The sugar content was evaluated using the PSA method, as previously described [39]. Briefly, 50 µL of the sample was added to wells in a 96-well plate, then 150 µL of 97% sulfuric acid was added, followed by 30 µL of 5% phenol. The plates were placed in an incubator for 5 min at 90 °C and then the absorbance was measured using a microplate reader at an OD of 490 nm.

### 2.5. Experimental Fish

The zebrafish (*Danio rerio*) used in the experiment were obtained from our facility at Ehime University. The experimental fish weighed 380–420 mg, with fork lengths of 32–34 mm; the fish were maintained at 25 ± 2 °C. The feeding regime was at satiation, twice daily (at 09.00 am and 5.00 pm), with a light/dark cycle of 14 h:10 h. All of the fish experiments were conducted in accordance with the Animal Experiments Regulations of Ehime University. The protocol was sanctioned by the Institutional Animal Care and Use Committee (IACUC) of Ehime University (Permit Number: 08K2-1). Surgical procedures (euthanized) and body measurements (anesthetized) were conducted using 2-phenoxyethanol at concentrations of 0.2% and 0.01%, respectively, to minimize suffering.

### 2.6. Preparation of Dipterose-BSF Diet

The composition of the zebrafish diet is shown in Table 1. Prior to pelletization with a cylindrical granulator (ABV-120L, Akira Kiko, Fukui, Japan), dry materials were thoroughly mixed, supplemented with fish oil, and then blended with water. The pellets were thoroughly air-dried after granulation for 1–2 days at 60 °C to a size of 2–3 mm. The dipterose-BSF diets were created by adding it to final concentrations of 0.01, 0.1, and 1 µg/g. As the quantities of the added dipterose-BSF were very small, their nutritional influence was considered to be insignificant.

### 2.7. Zebrafish mRNA Sequencing and Validation Using qRT-PCR

A total of 120 zebrafish were randomly divided into four aquariums (45 × 25 × 25 cm). Each tank received a different dipterose-BSF diet: 0.01, 0.1, or 1 µg/g, or no dipterose-BSF (control), for 14 days. The temperature was maintained at 25 °C with constant air filtering for 24 h, the light/dark cycle was 14 h:10 h, feed was provided to satiation at 09:00 am and 5:00 pm, and 30% of the water was changed from the total volume twice a week. After 14 days of dietary treatment, nine fish from each treatment group were euthanized and dissected to obtain liver and intestine samples for RNA extraction. RNA was isolated using ISOGEN II reagent, following the manufacturer’s protocol. To confirm RNA quality, the concentration and A280/A260 values of the extracted total RNA were measured using a trace spectrophotometer (NanoPhotometer P330, Implen, Munchen, Germany).

RNA-seq analysis was performed using four fish from each treatment, with comparisons made between the control group and the 0.1 µg/g dipterose-BSF treatment group. Samples were adjusted to a total RNA concentration and volume of at least 50 ng and 20 µL, respectively, and complete digestion of genomic DNA was performed using the TURBO DNA-freeTM Kit (Invitrogen, MA, USA). The concentration of the total RNA was measured using a QubitRNA HS Assay Kit (Thermo Fisher Scientific, MA, USA), and the sample qualities were confirmed using a Qsep100 DNA Fragment Analyzer and an RNA R1 Cartridge (BiOptic, Taiwan). The library DNA was set up using a KAPA Stranded mRNAseq kit (KAPA BIO SYSTEMS, MA, USA), according to the manufacturer’s instructions. The concentration of the prepared library DNA solution was assessed using Qubit and dsDNA HS Assay Kits (Thermo Fisher Scientific, MA, USA), and the quality of the library DNA was confirmed using a fragment analyzer and a dsDNA 915 Reagent Kit (Advanced Analytical Technologies, Ames, IA, USA).

Library DNA was cyclized using an MGIEasy Circularization Kit (MGI Tech Co., Ltd., Shenzhen, China) and then synthesized into DNA nanoballs (DNBs) using a DNBSEQ-G400RS High-throughput Sequencing Set (MGI Tech Co., Ltd.). DNB sequences were read using a NextSeq 500 (Illumina, CA, USA) at a sequencing depth of 2 × 76 bp. After eliminating adapter sequences from the obtained sequence data with Cutadapt (ver. 1.9.1) [40], Sickle (ver. 1.33) was used to eliminate bases with a quality score of less than 20 and paired reads with less than 30 bases. The filtered data reads were mapped using STAR (ver. 2.7.11b) to the reference sequence of *D*. *rerio* (GRCz11-GCA_000002035.4), to produce a file in the BAM format. The BAM file was indexed using Samtools (ver. 1.19.2) [41]. Reads corresponding to the gene regions of the reference sequence were counted using featureCounts (ver. 3.18) [42]. The TMM-edgeR-TMM pipeline was used to identify differentially expressed genes (DEGs) following normalization using the DEGES normalization technique in TCC (ver. 1.38) [43]. The thresholds for up- and downregulation were set at log2 fold-alterations in genes of 1 or −1 times with a false discovery rate (FDR) of <0.05 when compared with the control group. Multidimensional scaling, heatmap, and volcano plots were generated using the metaseqR2 v1.10 [44] and ggplot2 v3.4.3 [45] packages. ClusterProfiler (version 3.18.1) [46] was used to perform Gene Ontology (GO) enrichment and pathway enrichment analysis (PEA) on the DEGs. The Kyoto Encyclopedia of Genes and Genomes (KEGG) database (https://www.kegg.jp/ accessed on 3 January 2024) was also used for PEA.

Seven genes were chosen for quantitative reverse transcription polymerase chain reaction (qRT-PCR) using a PowerTrack SYBR Green Master Mix (Thermo Fisher Scientific, Vilnius, Lithuania) according to the manufacturer’s protocols, with a confirmation assay employed for the same number of sample sets (*n* = 5 for each group). A High-Capacity cDNA Reverse Transcript Kit (Thermo Fisher Scientific, Vilnius, Lithuania) was used to obtain first-strand complementary DNA (cDNA) from 500 ng of total RNA. The thermocycling process was carried out in 96-well white Multiplate PCR Plates (Bio-Rad Laboratories, Tokyo, Japan), utilizing a qRT-PCR detection system (Bio-Rad Laboratories) with the following conditions: 30 s at 95 °C, then 40 cycles of 5 s at 95 °C and 5 s at 55 °C. Following amplification, melting curve procedures were performed for each gene to verify that only a single product was amplified. The comparative threshold (CT) cycle approach established by Livak and Schmittgen (2001) [47] was used to quantify relative gene expression, with elongation factor 1-alpha (elfa) as an internal reference. Elfa was selected as the housekeeping gene for zebrafish as it has a low degree of variability under a wide range of conditions, e.g., during development, across tissue types, and following chemical treatments [48]. The primers used for qRT-PCR are shown in Table 2.

### 2.8. Challenge Test

For the challenge test, we used a bacterial agent (*Edwardsiella tarda*) obtained from the kidney of an infected red sea bream (*Pagrus major*) under Edwardsiellosis, which we obtained from Ainan, Ehime Prefecture, Japan. A preliminary investigation showed that the LD_50_ of *E. tarda* in zebrafish was 1 × 10^7^ CFU/mL or an optical density of 0.5 at 660 nm. The *E. tarda* for the challenge test was prepared in 500 mL of LB medium (inoculation ratio 10 µL/100 mL) and shaken at 170 rpm for 17–18 h in a shaker incubator at 28 °C. The *E. tarda* culture was then centrifuged at 6000 rpm at 4 °C for 5 min, washed twice with phosphate-buffered saline (PBS) (same volume as the medium), followed by a third addition of PBS and shaking the bottle until the *E. tarda* formed a homogenous mixture. The immersion for infection ratio was 9:1 between culture water and homogenized *E. tarda* in PBS for 8 h.

The challenge test was carried out to analyze the immunomodulatory response in zebrafish after the fish had been subjected to 14 days of dipterose-BSF dietary treatment. This involved 40 zebrafish that were randomly divided into four aquariums of dimensions 45 × 25 × 25 cm (*n* = 10 fish per tank). The challenge test was repeated three times. The aquariums were equipped with a submerged air filtration system, and the fish continued to be fed on the experimental diet. The fish were maintained at 25 ± 2 °C and observed until the day after 100% mortality occurred in the control group.

### 2.9. Statistical Analysis

The standard error of the mean (SEM) was used to present all of the data. The statistical analyses were carried out using EZR software version 1.64 (https://www.jichi.ac.jp/saitama-sct/SaitamaHP.files/statmed.html accessed on February 5 2024). The survival rate data were compared using the Kaplan–Meier method, further investigated using the log-rank test followed by Holm’s post hoc test (*p* < 0.05), while mRNA expression was analyzed using one-way analysis of variance (ANOVA) subjected to Tukey’s post hoc test (*p* < 0.05).

## 3. Results

### 3.1. Transcriptome Analysis Using RNA-Seq after Fish Had Received Dietary Dipterose-BSF

The challenge test showed that the optimum concentration of dipterose-BSF as a feed supplement for enhancing immunity was 0.1 µg/g. Therefore, the challenge test result was included as our first result, ahead of the RNA sequencing results and gene expression results. RNA-seq analysis was conducted to study transcriptome changes in the liver and intestine of zebrafish that had received dietary dipterose-BSF (0.1 µg/g) for 14 days compared with the control group. The raw reads resulted from RNA-seq ranged from 14,811,378 to 16,380,572, with average clean reads between 97.76% and 98.03% (Appendix A).

The clean reads were mapped using STAR (ver. 2.7.11) and compared with genome assembly GRCz11 (GCF_000002035.6) as a reference, resulting in an average successful mapping rate of 93.41% (Appendix A). We used the multidimensional scaling (MDS) method plot to assess the data and check for the possibility of bias and found there was little or no bias, as all samples were clustered in accordance with their respective treatment (Appendix A). Heatmap plots based on normalized counts were generated from the liver (Appendix A) and intestine (Appendix A) samples to analyze the pattern of DEGs.

### 3.2. Differentially Expressed Genes Induced by Dietary Dipterose-BSF

The DEGs were detected using a threshold value of log2 fold-change ≥ 1 with FDR < 0.05. Analysis of DEGs (TCC package, version 1.38) showed that 748 genes in the liver were differentially expressed between the control group (fed on the basal diet) compared with the group fed on a diet containing 0.1 µg/g dipterose-BSF. The DEGs comprised 369 upregulated genes and 379 downregulated genes. Using the same method, we found 698 DEGs in the intestine, including 381 upregulated and 317 downregulated genes (Figure 1).

### 3.3. Gene Ontology Enrichment Analysis of DEGs

Next, we used GO enrichment analysis to investigate the molecular mechanisms in the liver and intestines of zebrafish after they received dietary dipterose-BSF. The DEG results from the control and dietary dipterose-BSF groups were then compared by performing an over-representation analysis (ORA) using ClusterProfiler software. In the zebrafish intestine, ORA showed that five molecular functions were enriched (*p* < 0.05) following the dietary dipterose-BSF treatment: monooxygenase activity, iron ion binding, tubulin binding, heme binding, and cytokine receptor binding (Figure 2).

The dipterose-BSF diet impacted more GO terms in the liver of zebrafish compared with in the intestine (*p* < 0.05). There were twelve GO biological process (BP), four cellular component (CC), and two molecular function (MF) terms (Figure 3). Notably, macroautophagy and processes utilizing autophagic mechanisms were increased, combined with an increase in cytokine receptor binding in the liver, suggesting that dipterose-BSF provides a good foundation for adaptive immunity. This is because during self-degradation or autophagy processes, antigens can be processed then presented by the major histocompatibility complex (MHC) to immune effector cells [49]. Furthermore, one of the functions of cytokines is to regulate the ability of dendritic cells to present antigens and migrate to lymph nodes, which helps to trigger the adaptive immune response [50].

### 3.4. DEG Signaling Pathway Analysis Using KEGG Pathway Analysis

Evaluation of the signaling pathways of zebrafish liver and intestine samples was carried out using KEGG pathway analysis in ClusterProfiler software. The DEGs compared between the control and treatment groups were processed in the KEGG pathway database (https://www.kegg.jp/ accessed on 3 January 2024) and then analyzed to determine if there were any statistically significant difference terms (*p* < 0.05). In the liver, five signaling pathways were differentially increased: autophagy (dre04140), nucleotide metabolism (dre01232), PPAR signaling pathway (dre03320), foxO signaling pathway (dre04068), and protein processing in the endoplasmic reticulum (dre04141) (Table 3). At the same time, KEGG pathway analysis of the intestine showed that the ribosome biogenesis in eukaryotes signaling pathway (dre03008) was significantly increased (Table 3).

Figure 4 shows a schematic KEGG pathway of protein processing in endoplasmic reticulum (dre04141), which showed that stress-related genes were downregulated, such as Herp, HRD1, as well as HSP family members (HSP40, HSP 70, BiP, and HSP 90).

### 3.5. Immune- and Stress-Related DEG Determination

The determination and categorization of immune- and stress-related DEGs were conducted using research from the literature, the NCBI database, GO annotation, and the KEGG pathway database. Both functional categories were found to be potentially influenced in the zebrafish liver and intestine by dietary dipterose-BSF (Table 4). Further interpretation of the interaction and function of these genes is provided in the discussion.

### 3.6. RNA-Seq Validation Using qRT-PCR

RNA-seq results from the comparison of the control and dietary dipterose-BSF groups were validated using qRT-PCR. This verification was performed to ensure the confidence and accuracy of the RNA-seq results. The log2 fold-change obtained from RT-qPCR was compared with the RNA-seq results using Spearman’s correlation (coefficient ρ = 0.929 and *p* < 0.01) (Figure 5).

### 3.7. qRT-PCR Analysis of Immune- and Stress-Related Genes

The impacted genes obtained from the zebrafish intestine and liver after dipterose-BSF dietary treatment are having ‘bell-shaped’ dose–response with 0.1 µg/g group as the highest peak. They included nuclear factor kappa-light-chain-enhancer of activated B-cells 2 (NF-κΒ2), nuclear factor interleukin 3, member 5 (NFIL3-5), nicotinamide dinucleotide phosphate oxidase (NOX1), and interferon-induced protein with tetratricopeptide repeats 8 (IFIT8) in the intestine. While Figure 6 shows the relative expression of protectin (CD59), hepcidin antimicrobial peptide (HAMP), heme oxygenase (HO-1), and cells nuclear factor interleukin 3, member 5 (NFIL3-5) in the liver.

### 3.8. Challenge Test Using E. tarda

The challenge test was carried out to investigate the immune status of zebrafish after they were fed on dipterose-BSF for 14 days and their resistance to infection by pathogen (Figure 7A–C). The zebrafish were challenged with *E. tarda* and observed until the day after 100% mortality occurred in the control group. The results showed that mortality first appeared on the third day post-challenge in all replicates, with the highest survival rate in the 0.1 µg/g group, with values of 60%, 40%, and 10% shown in replications 1, 2, and 3, respectively. There were no significant differences between the control group and the other treatment groups. The challenge test showed that the optimum concentration of dipterose-BSF as a feed supplement for enhancing immunity was 0.1 µg/g. Therefore, we placed the challenge test result first, ahead of the RNA sequencing results and gene expression results.

## 4. Discussion

Previously, we purified multiple bioactive polysaccharides from various insects, including from melon fly larva (*Bactrocera cucurbitae*), which we named dipterose-BC [36]; Japanese oak pupae (*Antheraea yamamai*), named silkrose-AY [37]; silkmoth pupae (*Bombyx mori*), named silkrose-BM [38]; and dipterose-BSF from BSF larvae (*H*. *illucens*) [25]. The use of natural substances as immunomodulators in aquaculture has increased considerably, as the use of antibiotics is now widely known to have detrimental effects, both on the environment (leading to the spread of drug-resistance genes) [51] and on humans (resulting in increased severity of infections, which could lead to greater mortality) [52]. According to Ali et al. (2019) [25], dipterose-BSF stimulated Toll-like receptor 2 (TLR2) and TLR4, activating various downstream signaling molecules in RAW264.7 cells, and leading to the expression of numerous proinflammatory cytokines, including interleukin (IL)-1ß, IL-6, and tumor necrosis factor alpha (TNF-α). Dipterose-BSF thus functions more as an immunomodulator than as a bactericidal agent; this would help to reduce the likelihood of resistance genes emerging in the environment.

Dipterose-BSF has practical and economic advantages when used in aquaculture, as BSF can convert low-value organic wastes and by-products into edible sources of protein and fat for fish feed production [53,54]. In the long term, the use of BSF as a fishmeal substitute could reduce commercial feed prices and promote the welfare of fish farmers, especially small-scale fish farmers in low- and middle-income countries who produce freshwater fish.

### 4.1. Immune- and Stress-Related Genes with Their Respective Receptor Alterations Following Dietary Dipterose-BSF Treatment

Inflammation comprises a complex set of homeostatic mechanisms that involve the neurological, circulatory, and immunological systems and occurs in response to infection or organ injury. If an acute inflammatory response fails to remove a pathogen, the inflammatory process continues and takes on new characteristics [55]. In our study, we found that various immune- and stress-related genes and their respective receptors in the zebrafish liver and intestine were significantly influenced by dietary dipterose-BSF treatment. Pattern recognition receptors (PRRs) are a type of immunological sensor that play critical roles in identifying and responding to conserved patterns found in microorganisms; they include TLRs, RIG-I-like receptors (RLRs), and NOD-like receptors (NLRs), together with their downstream molecules, which have all been identified and characterized in teleosts [56]. Our results showed that TLR/NF-κB was upregulated in both the liver and intestines of zebrafish (Table 4); according to Aoki (2013) [57], this indicates that PAMPs recognize foreign molecules via PRRs. This finding is similar to those reported by Ali and colleagues (2019, 2022) [25,58], who discovered that dipteroses are detected by TLR2 and TLR4, eventually triggering immunomodulatory activities in RAW264.7 cells and Japanese medaka (*Oryzias latipes*). Furthermore, dipteroses have been shown to increase the expression of NFIL3 and NFIL3-5, which are vital upstream regulators of the NF-κB pathway [59].

An interesting finding in our present study was that the downregulated genes were dominated by antiviral genes, for instance, IRF10, a protein responsible for stimulating antiviral activity in virus-infected cells and also promoting MHC [60]. Presumably, this occurs because dipterose-BSF, which is classified as a polysaccharide, has a structure that resembles the bacterial LPS found in Gram-negative bacteria and can thus activate innate antimicrobial immune responses. Bacterial LPS is a cell wall component of Gram-negative bacteria that can act as a PAMP, enabling host cells to recognize a bacterial invasion and triggering innate immune responses [61].

There is evidence to suggest that adaptive immune responses could also be indirectly affected (upregulated) by dietary dipterose-BSF, such as increased expression of the CD83 molecule, which has been found to be expressed across numerous activated immune cells, including B and T lymphocytes [62]. Fish NF-κB combines T-cell receptor (TCR) and IL-17 signals to modulate ancestral T-cell immune responses to bacterial infection [63].

Regarding stress-related genes, we categorized them based on their association with reactive oxygen species (ROS). The results showed that dietary dipterose-BSF treatment did not cause oxidative stress (Table 4). There were 23 stress-related genes that were downregulated, compared with 12 stress-related genes that were upregulated, due to the increased expression of five serine-related genes. According to Yang and colleagues (2021) [64], serine stimulates glutathione synthesis, which reduces the production of ROS and subsequently suppresses immunological responses, preventing an overactive immune response. Thus, in this study, the increased expression of serine-related genes might act to limit the increases in the immune response caused by dipterose-BSF.

One interesting finding of this study is that dietary dipterose-BSF significantly suppresses the expression of heat shock proteins (HSPs), including HSP40, HSP70, and HSP90 (Figure 4). The accumulation of HSP families is well-known to be associated with the intensity of stress. Thus, these proteins have been regarded as a suitable biomarker for assessing animals’ reactions to environmental and physiological stressors [65]. Taken together, dipterose-BSF might have a stress-reducing effect in teleost, suggesting a promising alternative to tackle stress effects in teleost, such as during transportation or global temperature rise. Further analysis in this regard is needed to confirm these phenomena.

### 4.2. Selected Immune- and Stress-Related Gene Alterations Following Dietary Dipterose-BSF Treatment Using qRT-PCR

Teleost inflammation processes protect the host from infection and serve as a vital mechanism for the activation of subsequent tissue healing systems [66]. We found that several AMPs, NADPH oxidase, and nuclear factors and their respective receptors were affected significantly in zebrafish liver and intestine after dietary dipterose-BSF treatment, resulting in a ‘bell-shaped’ dose–response, with the 0.1 µg/g group showing the highest peak (Figure 5).

Immune- and stress-related genes whose expression was altered after receiving dietary dipterose-BSF included CD59, HAMP, HO-1, and NFIL-3 (liver), as well as NF-κΒ, NFIL-3, NOX1, and IFIT8 (intestine). Proinflammatory signaling reshapes the milieu within an injury/infection site, stimulates leukocyte recruitment, promotes antimicrobial mechanisms, and leads to the resolution of inflammation [66]. According to Sun et al. (2013) [67], CD59 (also known as protectin) in zebrafish regulates inflammation and protects cell walls and membranes from rupture. This protein is useful throughout all life stages in zebrafish for suppressing cell breakdown caused by viruses. HAMP, known as hepcidin in teleosts, has two different roles: regulating iron metabolism and antimicrobial activity [68]. Furthermore, Neves and colleagues (2015) [69] found that enhanced HAMP1 expression is a response to limit the iron available to pathogens by increasing iron retention and mobilization, while HAMP2 is associated with direct antimicrobial activity. The NF-κΒ signaling pathway is involved in several physiological processes, including inflammation, immunological responses, apoptosis, and cell proliferation and differentiation [70]. In this study, we found that the optimum NF-κΒ expression occurred in the 0.1 µg/g group (*p* < 0.05). We suspect this is because dipterose-BSF, as a polysaccharide, can activate NF-κΒ. More than 150 distinct stimuli can activate NF-κΒ, including bacterial LPS, proinflammatory cytokines such as TNF-α and IL-1, hormones, and mitogenic substances [71].

In addition to proinflammatory activities, we also found anti-inflammatory responses in zebrafish livers following exposure to dietary dipterose-BSF. We found that heme oxygenase (HO-1) expression was significantly higher in the 0.1 µg/g group. Anti-inflammatory processes in teleosts function to suppress immune responses [72], to prevent excessive inflammation and subsequent tissue damage. Based on this result, we presume that an optimum dose of dietary dipterose-BSF could promote immune homeostasis. In addition, we found that genes related to tissue regeneration, such as NFIL-3 [73] and NOX-1 [74], were also upregulated by the dietary inclusion of dipterose-BSF. NADPH oxidases, or NOXes, are enzymes that play an important role in the production of ROS following injury [75]. Increased levels of NFIL-3 mRNA in myeloid cells could also lead to tissue regeneration post injury and the overall development of the hematopoietic system of zebrafish [73]. Furthermore, zebrafish immune systems attempt to achieve homeostasis conditions, and Table 4 shows that several TNFs are upregulated both in the liver and intestine. According to Li et al. (2021) [76], TNFs are cytokines that play a crucial role in disease pathogenesis and homeostasis. Furthermore, TNFs are one of the most important types of proinflammatory cytokines involved in systemic inflammation and are required for effective innate and adaptive immune responses. These results suggest that dipterose-BSF can not only alter the inflammatory response but also plays a role in maintaining homeostasis and stimulating wound healing, both of which are vital for the survival of fish.

### 4.3. Challenge Test

We found that the optimum dose of dipterose-BSF to enhance zebrafish immunity was 0.1 µg/g. A higher dose of dipterose-BSF did not provide greater protection against infection (Figure 6). This is because high doses of polysaccharides can have adverse effects on a fish’s immune system/immunosuppression [77] or lead to immune fatigue [78]. In addition, Wu et al. (2023) [79] reported that a high concentration of polysaccharide (laminarin) can cause proteobacteria amplification in the intestine of largemouth bass. Qin et al. (2023) [80] also found that the majority of *Bacillus* genera, a beneficial bacterium in the intestine, declined to varying degrees following all polysaccharide (laminarin) treatments they tested; however, a low dose of polysaccharide was shown to increase *Lactobacillus* and *Klebsiella* abundance in the intestine of spotted seabass (*Lateolabrax maculatus*), promoting the growth of beneficial bacteria.

As mentioned above, dipterose-BSF stimulates the immune system of zebrafish at the level of gene expression. Although some of our challenge trials showed dipterose-BSF had a significant effect on disease resistance (Figure 7A–C), not all trials showed this, and even those trials did not show sufficiently high survival rates. These results may indicate that dipterose-BSF can increase disease resistance in fish, but not to the same extent that has been observed with silkrose-BM [38].

## 5. Conclusions

In conclusion, dietary dipterose-BSF, a polysaccharide derived from *H*. *illucens*, at a concentration of 0.1 µg/g, effectively increased immune- and stress-related gene expression in zebrafish and reduced *E*. *tarda* infection. Transcriptome analysis showed that dipterose-BSF can stimulate many host immune responses to help resist bacterial infection. This study highlights the potential of extracted polysaccharides as a means of disease control in aquaculture. Despite the fact that we thoroughly investigated the effect of dietary dipterose on zebrafish immune responses, further studies should be conducted to investigate the long-term dietary effects, together with other factors such as reproduction, growth, post-harvest quality, and, ultimately, production costs. This will help to determine the optimal usage of dipterose-BSF in commercial aquaculture production.

## Figures and Tables

**Figure 1 insects-15-00326-f001:**
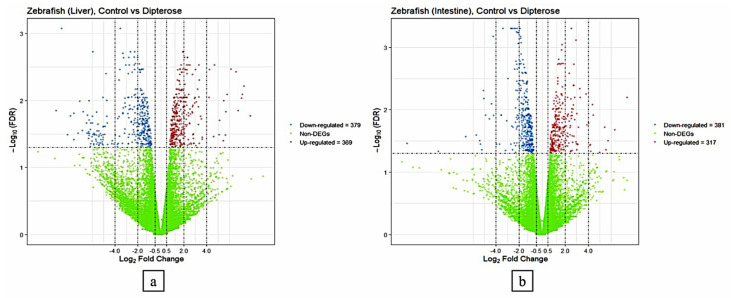
Transcriptional profiling analysis of the liver and intestine of zebrafish in the control group (fed on the basal diet) compared with the group fed on a diet containing 0.1 µg/g dipterose-BSF. (**a**) Expression patterns of differentially expressed genes (DEGs) in the livers of zebrafish in the control and dipterose-BSF groups. (**b**) Expression patterns of DEGs in the intestines of zebrafish in the control and dipterose-BSF groups. The DEGs were analyzed using the TCC package in R software.

**Figure 2 insects-15-00326-f002:**
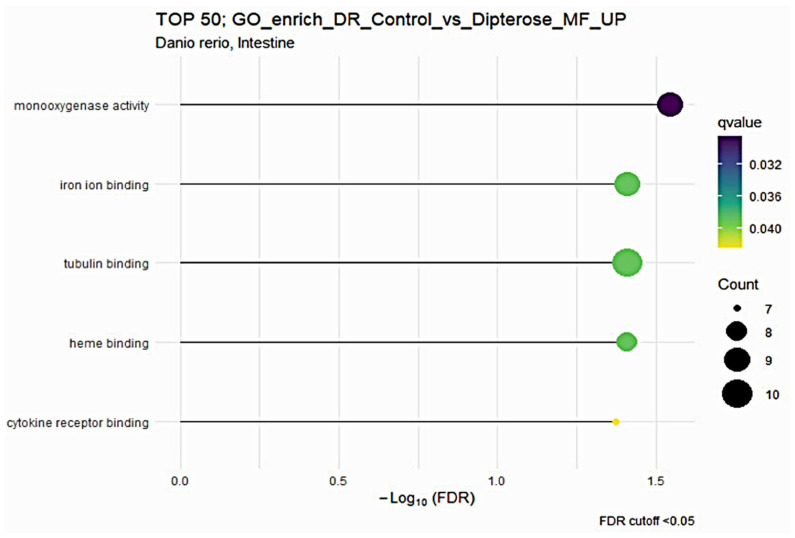
The Gene Ontology (GO) of molecular function (MF) terms in the zebrafish intestine based on DEGs generated from the comparison between the dipterose-BSF diet group and the control (basal diet) group.

**Figure 3 insects-15-00326-f003:**
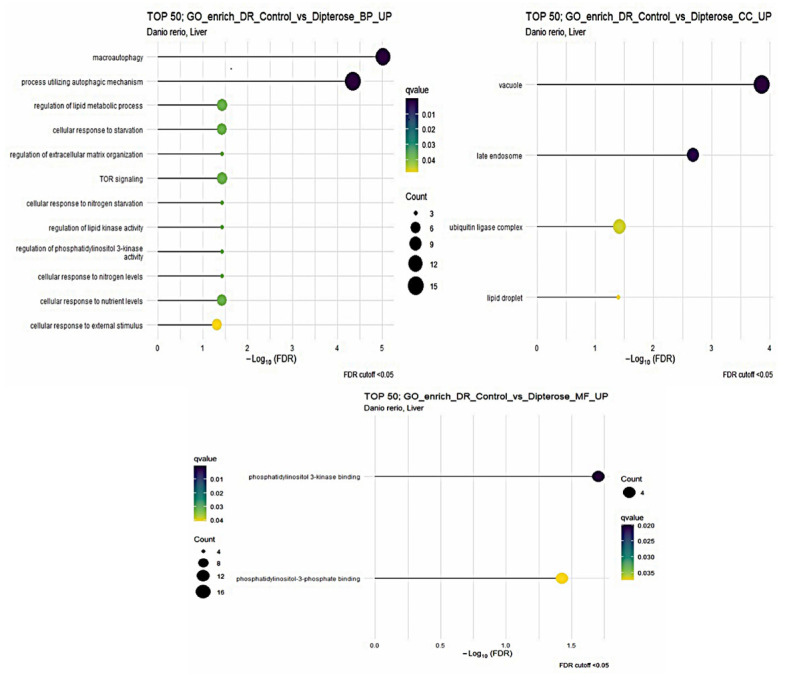
The Gene Ontology (GO) of biological process (BP), cellular component (CC), and molecular function (MF) terms in the zebrafish liver based on DEGs generated from the comparison between the dipterose-BSF diet group and the control (basal diet) group.

**Figure 4 insects-15-00326-f004:**
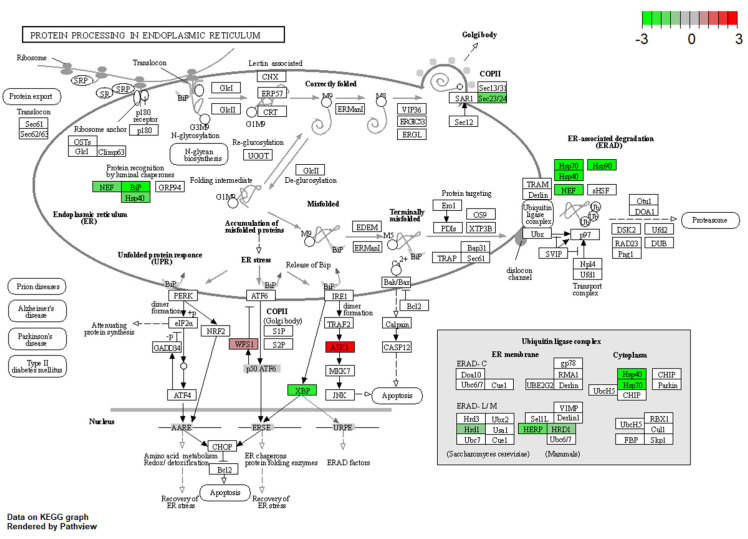
DEGs relevant to protein processing in endoplasmic reticulum, were generated from the comparison between the dipterose-BSF and the control (basal diet) groups. The green color indicates the downregulated gene, and the red color means the gene was upregulated.

**Figure 5 insects-15-00326-f005:**
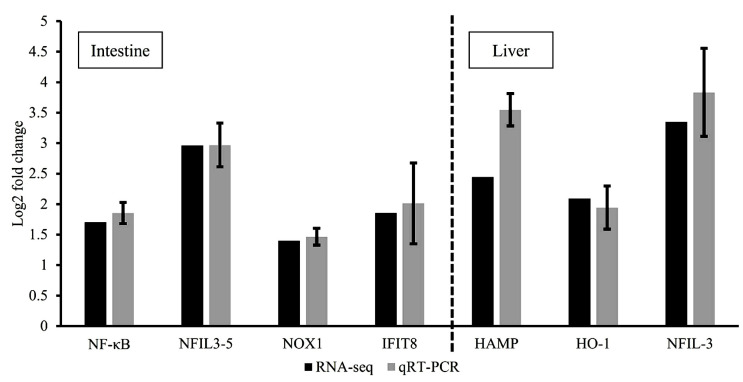
Validation of RNA-seq results using quantitative RT-PCR. Comparison of relative fold-changes between RNA-seq and qRT-PCR results in zebrafish intestines and liver. NF-κΒ = nuclear factor kappa-light-chain-enhancer of activated B cells; NFIL3-5 = nuclear factor, interleukin 3 regulated, member 5; NOX1 = NADPH oxidase 1; IFIT8 = IFN-induced protein with tetratricopeptide repeats 8; HAMP = hepcidin antimicrobial peptide; HO-1 = heme oxygenase 1; NFIL-3 = nuclear factor, interleukin 3 regulated. Values of qRT-PCR are shown as relative fold-changes between dipterose-BSF treatment and control groups to match the format of the RNA-seq results. Bars indicate the SEM of five samples (*n* = 5).

**Figure 6 insects-15-00326-f006:**
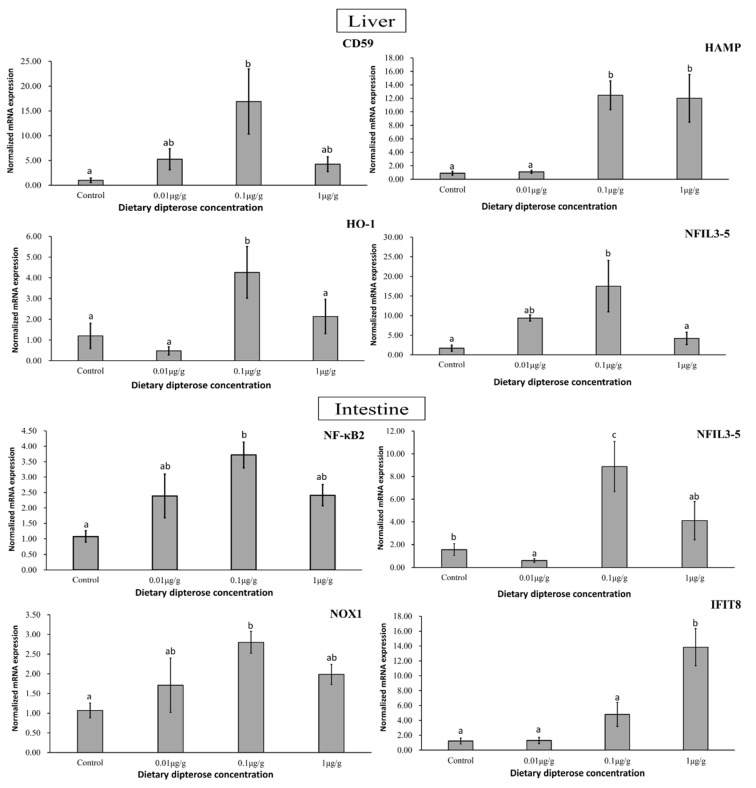
Expression patterns of immune- and stress-related genes in zebrafish liver and intestine following 14 days of dietary dipterose-BSF treatment using the 2^−ΔΔCT^ method (*n* = 5). Significance was investigated using a one-way analysis of variance (ANOVA) followed by Tukey’s post hoc test. Different letters indicate significance (*p* < 0.05).

**Figure 7 insects-15-00326-f007:**
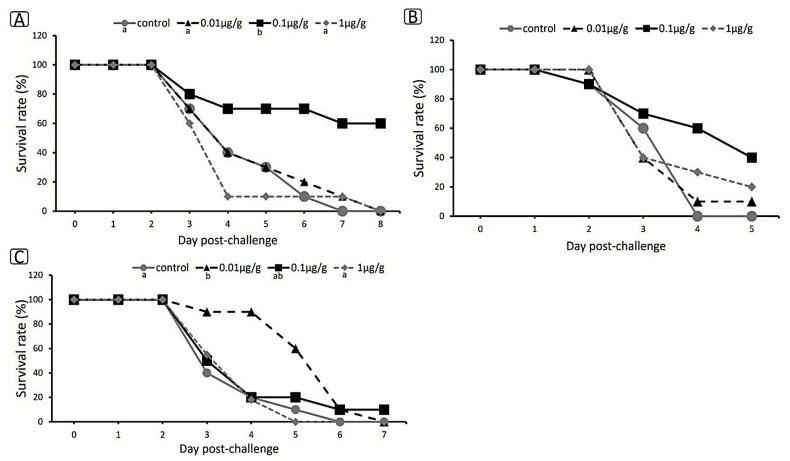
Survival rate of zebrafish after 14 days of treatment with dietary dipterose-BSF when challenged with *E. tarda* using the immersion method. (**A**) Replication 1. (**B**) Replication 2. (**C**) Replication 3. Data were compared using the Kaplan–Meier method, statistical significance was obtained with log-rank tests, and the Holm’s method was used for multiple pairwise comparisons (*p* < 0.05). Small letters indicate significance.

**Table 1 insects-15-00326-t001:** Zebrafish feed formulation.

Materials	Dry Weight (g)
Krill meal	100
Fish meal	600
Soybean meal	50
Wheat flour	93
Feed Thickeners (wheat starch)	60
Fish oil	50
Soy lecithin	10
Vitamin mixture *	8
Mineral mixture **	6
Choline chloride	1
Sodium dihydrogen phosphate	6
Potassium dihydrogen phosphate	6
Calcium lactate	10
Total	1000

* Vitamin mixture contained Vitamin A (2000 IU/g), Vitamin D3 (480 IU/g), Vitamin E (64 mg/g), Vitamin K3 (614.4 mg/g), Vitamin B1 (6.272 mg/g), Vitamin B2 (6.4 mg/g), Calcium D-pantothenate (15.84 mg/g), Nicotinic acid (14.4 mg/g), Vitamin B6 (4.752 mg/g), Biotin (0.32 mg/g), Folic acid (0.792 mg/g), Vitamin B12 (0.064 mg/g), 2-O-α-D-Glucopyranosyl-L-ascorbic acid (42 mg/g) and Inositol (78.4 mg/g). ** Mineral mixture contained Zinc sulphate (50.60 mg/kg), Magnesium sulphate (39.20 mg/kg), Ferrous sulphate (58.82 mg/kg), Ferrous fumarate (65.78 mg/kg), Manganese sulphate (17.46 mg/kg), Calcium lactate (224.88 mg/kg), Aluminium hydroxide (0.40 mg/kg), Calcium iodate (0.24 mg/kg), Cobalt chloride (0.09 mg/kg), Copper sulphate (39.80 mg/kg), and Peptide-iron (30 mg/kg).

**Table 2 insects-15-00326-t002:** Primers used in quantitative real-time PCR.

Primer Name	Primer Sequence	NCBI Accession Number	Efficiency (%)
Hepcidin Fw	TAACGTGTTTCTGGCTGCTG	NM_205583	110.4
Hepcidin Rv	GCCTTTATTGCGACAGCATT		
Heme oxygenase 1a Fw	CCACGTCAGAGCTGAAAACA	NM_001127516	105.2
Heme oxygenase 1a Rv	CCGAAGAAGTGCTCCAAGTC		
NADPH oxidase 1 Fw	TGCACATCCGCTCTGTTGGA	NM_001102387	104.1
NADPH oxidase 1 Rv	AGGCAAATGGGGTCACTCCA		
CD59 Fw	TGATGAAGGTTCTGCTGCTG	NM_001326385	120.6
CD59 Rv	GATGCACCTTCGGAAGTAGG		
NFKB2 Fw	AGGAGCCAAAGCAGAGAGGA	XM_005156814.4	110.3
NFKB2 Rv	AACCTCCACACGAGCATTGC		
Nfil 3–5 Fw	GAAAACAGGCGGTTTGTCAT	NM_001197058	106.2
Nfil 3–5 Rv	AGAGTGCCGAGGTAGGGATT		
Ifit8 Fw	AGAGCTAAAGCAGGCAGGCA	NM_001037565	100.2
Ifit8 Rv	GGACACTCGTCCCAGCATCA		

**Table 3 insects-15-00326-t003:** Significantly enriched KEGG (Kyoto Encyclopedia of Genes and Genomes) pathways (*p* < 0.05) analyzed by ClusterProfiler (ver. 3.18.1) using a comparison of DEGs between the control group and the dietary dipterose-BSF group.

No.	KEGG ID	Description	*p*-Value	Count
Liver				
1	dre04140	Autophagy–animal	8.57 × 10^−5^	20
2	dre01232	Nucleotide metabolism	0.000339	12
3	dre03320	PPAR signaling pathway	0.001278	10
4	dre04068	FoxO signaling pathway	0.001317	15
5	dre04141	Protein processing in endoplasmic reticulum	0.001396	16
Intestine				
1	dre03008	Ribosome biogenesis in eukaryotes	5.06 × 10^−16^	22

**Table 4 insects-15-00326-t004:** Full list of differentially expressed immune- and stress-related genes (|log2 fold-change| ≥ 1 and FDR < 0.05) and their expression ratios in the liver and intestine of zebrafish.

Gene Name	Gene Symbol	Gene ID	Fold log2 Change (FDR Value)	
			Liver	Intestine
Immune-related genes (upregulated)				
Nuclear factor, interleukin 3 regulated	NFIL3	ENSDARG00000042977	NE	2.549 (0.001)
Nuclear factor, interleukin 3 regulated, member 5	NFIL3-5	ENSDARG00000094965	3.351 (0.009)	2.961 (0.001)
Nuclear factor of kappa light polypeptide gene enhancer in B-cells 2	NFKB2	ENSDARG00000038687	1.779 (0.003)	1.703 (0.001)
Nuclear receptor subfamily 3, group C, member 1	NR3C1	ENSDARG00000025032	NS	0.925 (0.041)
Nuclear receptor subfamily 2, group C, member 2	NR2C2	ENSDARG00000042477	1.376 (0.018)	NS
TNF receptor-associated factor 4b	TRAF4	ENSDARG00000038964	NS	1.840 (0.002)
Tumor necrosis factor, alpha-induced protein 8-like 1	TNFAIP8L1	ENSDARG00000086457	NS	0.767 (0.045)
Tumor protein p53 inducible protein 11a	TP53I11a	ENSDARG00000069430	2.972 (0.009)	NS
TNF superfamily member 10, like	TNFSF10L	ENSDARG00000004196	NS	1.380 (0.006)
TNF superfamily member 10	TNFSF10	ENSDARG00000057241	NS	1.760 (0.008)
Microtubule-associated tumor suppressor 1a	MTUS1a	ENSDARG00000071562	NS	1.720 (0.002)
Filamin A interacting protein 1	FILIP1	ENSDARG00000078419	NS	2.361 (0.003)
Filamin A interacting protein 1	FILIP1	ENSDARG00000079634	2.092 (0.031)	NS
Platelet-derived growth factor alpha polypeptide b	PDGF-b	ENSDARG00000098578	NS	1.367 (0.002)
Interferon alpha inducible protein 27.3	IFI27.3	ENSDARG00000074217	NS	1.864 (0.040)
BCL2 family apoptosis regulator BOK a	BOKa	ENSDARG00000052129	NS	0.958 (0.019)
Mitogen-activated protein kinase 14a	MAPK14a	ENSDARG00000000857	NS	0.738 (0.029)
Mitogen-activated protein kinase 6	MAPK6	ENSDARG00000032103	NS	0.768 (0.042)
Mitogen-activated protein kinase 14b	MAPK14b	ENSDARG00000028721	1.405 (0.016)	NS
Mitogen-activated protein kinase kinase kinase 5	MAP3K5	ENSDARG00000005416	1.408 (0.009)	NS
Chemokine (C-X-C motif) ligand 18a, duplicate 1	CXCL18a.1	ENSDARG00000111840	NS	1.397 (0.010)
Chemokine (C-X-C motif) receptor 3, tandem duplicate 3	CXCR3.3	ENSDARG00000070669	1.394 (0.011)	NS
Atypical chemokine receptor 4b	ACKR4b	ENSDARG00000040133	1.067 (0.037)	NS
Thyroid hormone responsive	THRSP	ENSDARG00000099399	1.670 (0.017)	3.028 (0.012)
Hepcidin antimicrobial peptide	HAMP	ENSDARG00000102175	2.446 (0.017)	2.835 (0.006)
Caspase 6b.1, apoptosis-related cysteine peptidase	CASP6b.1	ENSDARG00000025608	NS	1.273 (0.017)
Caspase 6b.2, apoptosis-related cysteine peptidase	CASP6b.2	ENSDARG00000070368	NS	1.997 (0.007)
Protein regulator of cytokinesis 1a	PRC1a	ENSDARG00000100918	NS	1.888 (0.023)
Immunity-related GTPase family, f1	IRGF1	ENSDARG00000070774	1.805 (0.030)	NS
CD83 molecule	CD83	ENSDARG00000079553	NS	1.551 (0.036)
Bloodthirsty-related gene family, member 2	BTR02	ENSDARG00000052215	NS	0.926 (0.036)
Bloodthirsty-related gene family, member 25	BTR25	ENSDARG00000102018	NS	1.387 (0.038)
T cell immune regulator 1, ATPase H+ transporting V0 subunit a3b	TCIRG1b	ENSDARG00000105142	0.931 (0.037)	NS
T cell activation inhibitor, mitochondrial	TCIAM	ENSDARG00000079881	1.265 (0.032)	NS
Switching B cell complex subunit SWAP70b	SWAP70b	ENSDARG00000057286	0.908 (0.034)	NS
Serine active site containing 1	SERAC1	ENSDARG00000056121	1.387 (0.039)	NS
Serine/threonine kinase 11 interacting protein	STK11IP	ENSDARG00000070122	1.830 (0.004)	NS
Coiled-coil serine-rich protein 2	CCSER2	ENSDARG00000087749	1.609 (0.026)	NS
MAPK interacting serine/threonine kinase 1	MKNK1	ENSDARG00000018411	1.480 (0.035)	NS
Serine protease 35	PRS35	ENSDARG00000059081	4.638 (0.035)	NS
EI24 autophagy-associated transmembrane protein	EI24	ENSDARG00000053840	1.543 (0.009)	NS
Autophagy related 4A, cysteine peptidase	ATG4A	ENSDARG00000014531	0.998 (0.042)	NS
NBR1 autophagy cargo receptor a	NBR1a	ENSDARG00000077297	1.169 (0.042)	NS
Shiftless antiviral inhibitor of ribosomal frameshifting	SHFL	ENSDARG00000052176	1.436 (0.043)	NS
Leucine-rich repeat containing 8 VRAC subunit A	LRRC8A	ENSDARG00000052155	1.443 (0.025)	NS
Leucine-rich repeat containing 8 VRAC subunit Db	LRRC8DB	ENSDARG00000103840	1.025 (0.031)	NS
Leucine-rich repeats and immunoglobulin-like domains 2	LRIG2	ENSDARG00000078561	1.009 (0.026)	NS
Cysteine/histidine-rich 1	CYHR1	ENSDARG00000061057	1.574 (0.033)	NS
**Immune-related genes (downregulated)**				
Nuclear receptor corepressor 2	NCOR2	ENSDARG00000000966	NS	−0.814 (0.030)
Nuclear receptor subfamily 4, group A, member 2b	NR4a2b	ENSDARG00000044532	NS	−1.628 (0.035)
Tumor necrosis factor receptor superfamily, member 1a	TNFRSF1a	ENSDARG00000018569	−1.021 (0.043)	NS
Fibrinogen-like 2a	FGL2a	ENSDARG00000019861	−1.070 (0.035)	−1.242 (0.025)
Interferon-related developmental regulator 2	IFRD2	ENSDARG00000036811	NS	−1.195 (0.019)
Interferon regulatory factor 10	IRF10	ENSDARG00000027658	NS	−0.877 (0.043)
Major histocompatibility complex class II DGB gene	MHC-II DGB	ENSDARG00000104635	−7.597 (0.039)	−11.690 (0.035)
RNA polymerase I and III subunit C	POLR1C	ENSDARG00000039400	NS	−2.010 (0.002)
3′-phosphoadenosine 5′-phosphosulfate synthase 2a	PAPSS2a	ENSDARG00000071021	NS	−0.778 (0.045)
Tetratricopeptide repeat domain 4	TTC4	ENSDARG00000044405	NS	−2.028 (0.001)
Thyroglobulin	TG	ENSDARG00000020084	−8.126 (0.033)	−4.222 (0.001)
SWI/SNF-related, matrix-associated actin-dependent regulator of chromatin, subfamily a, containing DEAD/H box 1 a	SMARCAD1a	ENSDARG00000014041	−1.460 (0.038)	NS
DEAD (Asp-Glu-Ala-Asp) box polypeptide 4	DDX4	ENSDARG00000014373	−9.133 (0.014)	NS
DEAD (Asp-Glu-Ala-Asp) box polypeptide 18	DDX18	ENSDARG00000030789	NS	−1.204 (0.009)
DEAD (Asp-Glu-Ala-Asp) box polypeptide 54	DDX54	ENSDARG00000105286	NS	−1.099 (0.019)
DEAD-box helicase 24	DDX 24	ENSDARG00000104708	NS	−1.139 (0.018)
DEAD-box helicase 31	DDX31	ENSDARG00000035507	NS	−1.357 (0.036)
Caspase 22, apoptosis-related cysteine peptidase	CASP22	ENSDARG00000091926	NS	−1.072 (0.015)
Bloodthirsty-related gene family, member 12	BTR12	ENSDARG00000051809	NS	−1.590 (0.023)
Apoptosis inhibitor 5	API5	ENSDARG00000033597	−0.963 (0.034)	−0.734 (0.044)
Synovial apoptosis inhibitor 1, synoviolin	SYVN1	ENSDARG00000017842	−1.195 (0.031)	NS
MAPK-regulated corepressor-interacting protein 2	MCRIP2	ENSDARG00000061256	−0.989 (0.048)	NS
Basic leucine zipper and W2 domains 1b	BZW1b	ENSDARG00000099148	−2.285 (0.002)	−1.930 (0.001)
Serine palmitoyltransferase, long chain base subunit 2b	SPTLC2b	ENSDARG00000074287	−1.777 (0.003)	−1.749 (0.006)
Phosphatidylserine synthase 2	PTDSS2	ENSDARG00000101018	−1.154 (0.031)	NS
Immunoglobulin superfamily DCC subclass member 4	IGDCC4	ENSDARG00000076919	−0.844 (0.037)	NS
Leucine-rich repeat containing 59	LRRC59	ENSDARG00000071426	−2.216 (0.009)	NS
Homocysteine-inducible, endoplasmic reticulum stress-inducible, ubiquitin-like domain member 1	HERPUD1	ENSDARG00000024314	−1.462 (0.017)	NS
**Stress-related genes (upregulated)**				
Heme oxygenase 1a	HMOX1a	ENSDARG00000027529	2.090 (0.046)	2.287 (0.012)
Tet methylcytosine dioxygenase 2	TET2	ENSDARG00000076928	NS	0.878 (0.036)
Hemopexin a	HPXa	ENSDARG00000012609	NS	1.624 (0.044)
NADPH oxidase 1	NOX1	ENSDARG00000087574	NS	1.401 (0.046)
NADH:ubiquinone oxidoreductase core subunit S2	NDUFS2	ENSDARG00000007526	NS	0.770 (0.036)
Dehydrogenase/reductase (SDR family) member 3b	DHRS3b	ENSDARG00000044803	NS	0.929 (0.019)
Glutathione S-transferase theta 1b	GSTT1b	ENSDARG00000017388	NS	1.248 (0.019)
Glutathione S-transferase rho	GSTR	ENSDARG00000042620	NS	0.978 (0.035)
Pirin	PIR	ENSDARG00000056638	1.612 (0.005)	NS
Mannosidase, endo-alpha	MANEa	ENSDARG00000001898	1.014 (0.019)	0.835 (0.023)
Xanthine dehydrogenase	XDH	ENSDARG00000055240	1.096 (0.023)	1.365 (0.014)
Egl-9 family hypoxia-inducible factor 1b	EGLN1B	ENSDARG00000004632	1.330 (0.025)	NS
**Stress-related genes (downregulated)**				
Hypoxia upregulated 1	HYOU1	ENSDARG00000013670	−1.567 (0.044)	−1.388 (0.009)
Heat shock protein 4a	HSPA4a	ENSDARG00000004754	−2.506 (0.004)	−2.130 (0.001)
Heat shock protein 5	HSPA5	ENSDARG00000103846	−3.629 (0.005)	−2.133 (0.012)
DnaJ heat shock protein family (Hsp40) member C21	DNAJC21	ENSDARG00000105195	−1.899 (0.025)	−2.376 (0.002)
DnaJ heat shock protein family (Hsp40) member B1a	DNAJB1a	ENSDARG00000099383	−2.193 (0.008)	NS
DnaJ heat shock protein family (Hsp40) member B1B	DNAJB1b	ENSDARG00000041394	−2.451 (0.032)	NS
DnaJ heat shock protein family (Hsp40) member A1	DNAJA1	ENSDARG00000030972	−2.553 (0.021)	NS
HSPA (heat shock 70kDa) binding protein, cytoplasmic cochaperone 1	HSPBP1	ENSDARG00000102937	−2.272 (0.003)	−2.241 (0.005)
Heat shock cognate 70-kd protein, tandem duplicate 3	HSP70.3	ENSDARG00000021924	−3.265 (0.013)	−4.385 (0.008)
Heat shock transcription factor family member 5	HSF5	ENSDARG00000104686	−4.947 (0.025)	NS
AHA1, activator of heat shock protein ATPase homolog 1b	AHSA1b	ENSDARG00000100317	−1.528 (0.023)	−1.050 (0.013)
Heat shock protein 90, alpha (cytosolic), class A member 1, tandem duplicate 2	HSP90aa1.2	ENSDARG00000024746	−4.796 (0.004)	−2.299 (0.016)
Heat shock protein 9	HSPA9	ENSDARG00000003035	NS	−1.064 (0.018)
Nitric oxide-associated 1	NOA1	ENSDARG00000102934	NS	−1.494 (0.022)
Butyrobetaine (gamma), 2-oxoglutarate dioxygenase	BBOX1	ENSDARG00000036135	NS	−1.166 (0.010)
Oxidative stress responsive kinase 1a	OXR1a	ENSDARG00000034189	NS	−0.801 (0.045)
Glutathione S-transferase mu tandem duplicate 3	GSTM.3	ENSDARG00000088116	NS	−1.741 (0.018)
Lactase	LCT	ENSDARG00000092404	NS	−1.564 (0.036)
Phosphoenolpyruvate carboxykinase 1	PCK1	ENSDARG00000013522	−3.313 (0.002)	NS
Oxoglutarate dehydrogenase	OGDH	ENSDARG00000103428	−1.670 (0.003)	NS
L-threonine dehydrogenase	TDH	ENSDARG00000002745	−1.328 (0.012)	NS
UDP-glucose 6-dehydrogenase	UGDH	ENSDARG00000019838	−1.367 (0.016)	NS
Methylenetetrahydrofolate dehydrogenase	MTHFD2	ENSDARG00000098646	−1.600 (0.019)	NS

NE = not expressed; NS = not significant.

## Data Availability

The supporting data regarding this research are available from the corresponding author upon reasonable request.

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
