# Peer review of "Dietary Black Soldier Fly (Hermetia illucens)—Dipterose-BSF—Enhanced Zebrafish Innate Immunity Gene Expression and Resistance to Edwardsiella tarda Infection"

_insects, 2024, doi:10.3390/insects15050326_

Round 1
Reviewer 1 Report
Comments and Suggestions for Authors
The manuscript describes successfully the immunomodulating effect of dietary BSF extracts in zebrafish and their potential to increase disease resistance. The scientific method is overall well described. It would have been interesting to also look at immunological parameters to check if function is also modulated rather than just gene expression. Introduction can be improved by listing some insect meals and extracts with activity on animal immune system and resistance to diseases.The bacterial challenge should have been planned to get 50-60% mortality in the control diet so that relative protection survival would have been more reproducible. Some figures are very blur and cannot be read correctly. The discussion still needs some work to be more clear and logical.
L53: Insects can be used in animal feed not only as protein alternatives but also as immunomodulators capable of increasing resistance to diseases (Gasco, et al. 2021. Beyond the protein concept: Health aspects of using edible insects on animals. Journal of Insects as Food and Feed, 7 (5): 715 - 741).
L138: 106
L144: what concentration of sulfuric acid was used?
L163: what size had the pellets?
L182: rephrase please: constant air filtering … 30% of the water was changed twice a week.?
L256 & 396 rephrase please: The fish were …observed until the day after 100% mortality was observed in the control group
Fig. 2 , 3, 4 are very blur and cannot be read.
L256 & Fig.7 the LD50 should be used for the bacterial challenge to avoid 100% mortality in control group. It is always better to get around 50-60% mortality in the control group if possible. The reproducibility of the 3 challenge trials may have been improved this way.
L418: It is not the antibiotic toxicity which leads to increased mortality in human but the associated increased severity of infection which could lead to higher mortality
L422: It is not very clear why : ‘Dipterose-BSF thus functions more as an immunomodulator than as a bactericidal agent; this would help to reduce the likelihood of resistance genes emerging in the environment. ‘ please explain.
L447: ‘dipteroses are correlated with TLR2 and TLR4’, do you mean with their expression?
L452-458: a bit confusing: you write that the anti-viral response is downregulated but then you write about bacterial LPS. How is this related?
L462: TNF is not involved in homeaostasis on its own. It is a pro-inflammatory cytokine working in combination with anti-inflammatory cytokines towards homeostasis.
L551: it did not prevent E tarda infection. At best, it reduced E.tarda induced-mortality.
L553: you used an extracted polysaccharides and conclude about insect meal. Do not generalise your results to insect meal.
Comments on the Quality of English Language
The quality of the English is good overall. Some slight improvements may bring clarity to some parts.
Please rephrase L144, 182, 256, 396.
Author Response
The manuscript describes successfully the immunomodulating effect of dietary BSF extracts in zebrafish and their potential to increase disease resistance. The scientific method is overall well described. It would have been interesting to also look at immunological parameters to check if function is also modulated rather than just gene expression. Introduction can be improved by listing some insect meals and extracts with activity on animal immune system and resistance to diseases.The bacterial challenge should have been planned to get 50-60% mortality in the control diet so that relative protection survival would have been more reproducible. Some figures are very blur and cannot be read correctly. The discussion still needs some work to be more clear and logical.
Answer: We are truly appreciate for the excellent suggestions. We added lists of insect meals and its derivatives which used in aquaculture. As for discussion chapter that need more clear and logical sentences, we did some adjustments as stated by the reviewer.
L53: Insects can be used in animal feed not only as protein alternatives but also as immunomodulators capable of increasing resistance to diseases (Gasco, et al. 2021. Beyond the protein concept: Health aspects of using edible insects on animals. Journal of Insects as Food and Feed, 7 (5): 715 - 741).
Answer: Thank you for the insight, we added the reference into the manuscript.
L138: 106
Answer: Thank you for the correction. Done.
L144: what concentration of sulfuric acid was used?
Answer: Concentration of sulfuric acid is 97% and added on the manuscript.
L163: what size had the pellets?
Answer: The pellets size are 2-3 mm, and added into the manuscript.
L182: rephrase please: constant air filtering … 30% of the water was changed twice a week.?
Answer: Rephrasing done, yes 30% of total volume was done twice every week to keep the water quality in optimum range for the zebrafish.
L256 & 396 rephrase please: The fish were …observed until the day after 100% mortality was observed in the control group
Answer: Rephrasing done.
Fig. 2 , 3, 4 are very blur and cannot be read.
Answer: Done. We used adobe express to increase the font brightness and make the image dpi higher than before, we are truly sorry if the results are far from expectation, this is our maximum knowledge regarding images editing.
L256 & Fig.7 the LD50 should be used for the bacterial challenge to avoid 100% mortality in control group. It is always better to get around 50-60% mortality in the control group if possible. The reproducibility of the 3 challenge trials may have been improved this way.
Answer: We sincerely appreciate your advice and suggestion. We will consider to do challenge test until the control group reached 50-60% mortality.
L418: It is not the antibiotic toxicity which leads to increased mortality in human but the associated increased severity of infection which could lead to higher mortality
Answer: We are highly appreciate for the suggestion, rephrasing done.
L422: It is not very clear why : ‘Dipterose-BSF thus functions more as an immunomodulator than as a bactericidal agent; this would help to reduce the likelihood of resistance genes emerging in the environment. ‘ please explain.
Answer: We are sorry if this sentence is not clear. Bactericidal agent in this sentence is referring to antibiotics, as mentioned several sentences before.
L447: ‘dipteroses are correlated with TLR2 and TLR4’, do you mean with their expression?
Answer: We did not mean expression. Dipteroses are recognized by TLR2 and TLR4 proteins as their pathogen recognition receptor (PRR). The activation of TLR2/4 thus facilitates the activation of several downstream signaling molecules, including critical roles in releasing NF-κB into the nucleus, triggering the induction of innate immune system.
We are sorry if the sentence is not clear. We corrected the sentence to give clarity to the reader.
L452-458: a bit confusing: you write that the anti-viral response is downregulated but then you write about bacterial LPS. How is this related?
Answer: To our limited knowledge, the polysaccharide having similar structure with the bacterial cell wall (LPS). Thus, we assumed anti-viral responses were downregulated because other immune responses were recorded significantly upregulated. And also it might be a process to achieve homeostasis.
L462: TNF is not involved in homeaostasis on its own. It is a pro-inflammatory cytokine working in combination with anti-inflammatory cytokines towards homeostasis.
Answer: Thank you for the excellent recommendations, we moved the discussion about homeostasis together in the next sub-chapter (4.2) after anti-inflammatory discussions.
L551: it did not prevent E tarda infection. At best, it reduced E.tarda induced-mortality.
Answer: Thank you for the suggestion, we changed ‘prevent’ to ‘reduced’.
L553: you used an extracted polysaccharides and conclude about insect meal. Do not generalise your results to insect meal.
Answer: Thank you for the suggestion, we changed ‘insect meal’ to ‘extracted polysaccharides’.
Comments on the Quality of English Language
The quality of the English is good overall. Some slight improvements may bring clarity to some parts.
Please rephrase L144, 182, 256, 396.
Answer: The manuscript was fully edited by a commercial English editing service (Cambridge English Correction Service Inc.), including the parts that were pointed out.
Reviewer 2 Report
Comments and Suggestions for Authors
The MS titled “Dietary black soldier fly (Hermetia illucens) dipterose-BSF enhanced zebrafish innate immunity gene expression and resistance to Edwardsiella tarda infection” is an interesting study. The results could be helpful to elucidation of active components contained in Dietary black soldier fly (Hermetia illucens). In general, this manuscript was well-written, but some issues should be clearifed:
1. The final dipterose-BSF concentrations was 0.01, 0.1, and 1 µg/g. These levels were extremely low. Therefore, one of my major concerns is that how do you evenly mix the very low levels of dipterose-BSF into the feed. In my experience, it is very difficult.
2. Line 180, do you have replicate tanks? Usually for feeding trials, we use triplicate tanks for each dietary treatment.
3. Line 222, why do you choose these seven genes for qPCR?
4. “2.8. Challenge Test”, Do you have a preliminary experiment for the determination of the most suitable challenge method?
5. “3. Results” why not measure some immune parameters? The phenotypic parameters are useful to indicate the immune status.
6. Line 270, and line 401-402, since it was the challenge test which showed that the optimum concentration of dipterose-BSF as a feed supplement to enhance immunity is 0.1 µg/g. Therefore, we’d better put the challenge test result as the first result, ahead of other RNA sequencing results or gene expression results.
7. There are mistakes in the caption of figure 1: b) Expression patterns of DEGs in the livers of zebrafish in the control and dipterose-BSF groups. The “liver” here should be “intestine”.
8. All the figures are blurred.
9. Please add P values for each gene in Table 6.
10. Figure 5, As the authors mentioned: “Values of qRT-PCR are expressed as the gene expression ratio between dipterose-BSF treatment and control groups after normalization using β-actin as an endogenous control gene using the 2–ΔΔCT method”,the gene expression in the control group should be normalized to be 1?
11. Line 222-233, how were the quality of primers assessed?
Comments on the Quality of English Language
Better be revised by native speakers.
Author Response
The MS titled “Dietary black soldier fly (Hermetia illucens) dipterose-BSF enhanced zebrafish innate immunity gene expression and resistance to Edwardsiella tarda infection” is an interesting study. The results could be helpful to elucidation of active components contained in Dietary black soldier fly (Hermetia illucens). In general, this manuscript was well-written, but some issues should be clearifed:
- The final dipterose-BSF concentrations was 0.01, 0.1, and 1 µg/g. These levels were extremely low. Therefore, one of my major concerns is that how do you evenly mix the very low levels of dipterose-BSF into the feed. In my experience, it is very difficult.
Answer: We are greatly appreciate your concern regarding how to evenly mixed the dipterose-BSF into the feed. We dissolved the freeze-dry form of the dipterose-BSF with distilled water which later be used to mixed all pellet ingredients. For thorough mixing, we manually mixed the feed using hands for 15-20 minutes.
- Line 180, do you have replicate tanks? Usually for feeding trials, we use triplicate tanks for each dietary treatment.
Answer: We used individual replicates for the RNA sequencing and qRT-PCR, considering the analysis costs. We maintained 30 fish for each treatment, and took 5 fish (n=5) randomly from each treatment to analyze. Since it was the challenge test which showed that the optimum concentration of dipterose-BSF as a feed supplement to enhance immunity is 0.1 µg/g. Therefore, we’d better put the challenge test result as the first result, ahead of other RNA sequencing results or gene expression results.
While we used triplicates for challenge test because we used 10 fish for each treatment.
- Line 222, why do you choose these seven genes for qPCR?
Answer: Because we wanted to investigate immune- and stress- related genes. Those seven genes represent pro-inflammatory and anti-inflammatory for the immune-related genes, also ROS production for the stress-related genes.
- “2.8. Challenge Test”, Do you have a preliminary experiment for the determination of the most suitable challenge method?
Answer: Yes, we have. Because this current study is a continuation with previous research, so we already have several data regarding the most suitable challenge test method.
- “3. Results” why not measure some immune parameters? The phenotypic parameters are useful to indicate the immune status.
Answer: Thank you for your great suggestions, we will pay huge consideration to put phenotypic parameters in our next challenge study. As for this study, after long discussions we perceived that immune parameters are sufficiently described by the genes expression and by the zebrafish survival rate.
- Line 270, and line 401-402, since it was the challenge test which showed that the optimum concentration of dipterose-BSF as a feed supplement to enhance immunity is 0.1 µg/g. Therefore, we’d better put the challenge test result as the first result, ahead of other RNA sequencing results or gene expression results.
Answer: Thank you very much for the correction. We changed it as instructed.
- There are mistakes in the caption of figure 1: b) Expression patterns of DEGs in the livers of zebrafish in the control and dipterose-BSF groups. The “liver” here should be “intestine”.
Answer: We are truly sorry, we missed that. Thank you for the correction.
- All the figures are blurred.
Answer: We done the utmost best to enhanced the images quality using dpi enhancer and adobe express. We are truly sorry if the results are still far from expectation, this is our maximum knowledge regarding images editing.
- Please add P values for each gene in Table 6.
Answer: We used FDR (adjusted p-value) to mark DEG or not DEG. And we added the FDR value to each gene in the table 6.
- Figure 5, As the authors mentioned: “Values of qRT-PCR are expressed as the gene expression ratio between dipterose-BSF treatment and control groups after normalization using β-actin as an endogenous control gene using the 2–ΔΔCT method”,the gene expression in the control group should be normalized to be 1?
Answer: We are truly sorry that we made a mistake in the caption of Figure 5. We used log2 fold-change between Control and Experimental groups rather than relative expression to match the format of RNA-sequencing results, enabling us to compare the RNA-seq and qPCR analysis results. The relative expression ratio for the genes mentioned in Figure 5 can be seen in Figure 6. To give clarity to the reader, we have changed the caption.
- Line 222-233, how were the quality of primers assessed?
Answer: To asses the quality of primer, we conducted primer efficiencies test on the primer used in qPCR. Briefly, 4 fold-dillution concentration series of pooled control sample was created and then performed RT-qPCR for each primer. The calculation of qPCR efficiency was then conducted using BioRad CFX Manager Software (ver. 3.1.), utilizing the standard formula of PCR efficiency (Pfaffl method), where (E) = 10(–1/slope) and % efficiency = (E – 1) × 100%. Primers with efficiency between 90 and 110% were considerably suitable for further RT-qPCR analysis. The corresponding information regarding the primer efficiency can be seen now in Table 2.
Comments on the Quality of English Language
Better be revised by native speakers.
Answer: The manuscript was fully edited by a commercial English editing service (Cambridge English Correction Service Inc.).
Round 2
Reviewer 2 Report
Comments and Suggestions for Authors
I have no other comments.
翻译
搜索
复制